# Plastic Debris in Nests of Two Water Bird Species Breeding on Inland Saline Lakes in a Mediterranean Biosphere Reserve

**DOI:** 10.3390/ani12223222

**Published:** 2022-11-21

**Authors:** Álvaro Luna, José A. Gil-Delgado, Edgar Bernat-Ponce

**Affiliations:** 1Department of Health Sciences, Faculty of Biomedical and Health Sciences, Universidad Europea de Madrid, 28670 Madrid, Spain; 2Department of Microbiology and Ecology, Cavanilles Institute of Biodiversity and Evolutionary Biology, University of Valencia, c/Catedrático José Beltrán, 2, 46980 Paterna, Spain; 3Faculty of Health Sciences, Universidad Europea de Valencia, Paseo de la Alameda, 7, 46010 Valencia, Spain

**Keywords:** anthropogenic waste, birds, macroplastics, pollution, gull-billed tern, black-winged stilt

## Abstract

**Simple Summary:**

Plastic pollution has become one of the main emerging ecological problems that has gained growing public and scientific interest in recent years. In this work, we detected the presence of plastics and other synthetic debris in a less studied ecosystem: inland wetlands. There, we confirmed the presence of anthropogenic materials in the nests of two bird species, the gull-billed tern (*Gelochelidon nilotica*) and the black-winged stilt (*Himantopus himantopus*), in a biosphere reserve. The detected debris probably originated from human activities (agriculture and domestic waste) and was probably dispersed by wind and water. Although no damage to the studied species was detected, the situation requires monitoring, which must extend to other wetland points and other species. Our paper records the spread of plastic pollution through wetland ecosystems and its interaction with unexpected wildlife.

**Abstract:**

Despite more studies being carried out to know the impacts associated with plastic debris and much effort being spent on marine ecosystems, the impacts of plastics on terrestrial and freshwater species remain largely unknown. Here, we explored the presence of anthropogenic materials in nests of two wader species, the gull-billed tern (*Gelochelidon nilotica*) and the black-winged stilt (*Himantopus himantopus*), breeding on the inland salt lakes in the “La Mancha Húmeda” Biosphere Reserve, Central Spain. We revealed the presence of anthropogenic debris, mainly macroplastics (>5 mm), in 2.4% and 12.5% of the sampled nests of the gull-billed tern and the black-winged stilt, respectively. The fragments found in nests ranged from 8 mm to 257 mm for the gull-billed tern and from 7 mm to 19 mm for the black-winged stilt. This debris showed no clear pattern of color or size and probably originated both in the agricultural activities in the surroundings and domestic refuse. Although we did not detect any pernicious impacts on adults or chicks (e.g., entangled, injured, or dead individuals), the presence of plastics and other human waste directly placed in nests located in a protected area should warn us about the ubiquity of these pollutants, and the endocrine and immunological effects, among others, that may reduce the recruitment of new animals to the population should be assessed.

## 1. Introduction

The accumulation of plastic waste in natural ecosystems has been recognized as a concerning threat for ecosystems all around the world [1,2,3]. Plastic pollution has become globally widespread and has reached oceans and seas (from the surface to the depths) [4,5,6], rivers, and lakes [7,8] as well as terrestrial ecosystems, including the poles, alpine glaciers, and high mountain peaks [9,10,11,12,13]. Many studies have quantified a wide variety of environmental impacts associated with plastics. For example, the following have been confirmed: their ingestion by 701 marine species and entanglement by another 354 [14]; their effect on the soil properties of agricultural areas [15]; their role as vectors for the spread of invasive species and pathogens [16], and impacts on human well-being and the economy [17]. The association between plastic debris and organisms is shaped by plastic properties (type, size, and color), but it also depends on each species and the habitat. In seabirds, there is evidence for the ingestion of hard, white or gray, and round plastics, while plastics used as materials to build nests have a different composition and are bright in color [18]. Similarly, there is evidence of more use of plastic strings in nests of great grey shrikes (*Lanius excubitor*) that breed on farmland compared to conspecifics inhabiting more naturalized areas [19].

Despite the growing number of studies that focus on ecological and environmental aspects related to plastic pollution [20], research carried out in marine environments clearly dominates, while relevant studies about many freshwater and terrestrial ecosystems and their associated species are lacking [21,22,23,24]. However, in recent years some studies have confirmed the interaction of plastics with birds inhabiting rivers and inland wetlands. For example, microplastics have been found in pellets of the common kingfisher (*Alcedo atthis*) in Italy [25], in the feces and feathers of waterfowl species in South Africa [26], and inside the digestive systems of chicks of double-crested cormorants (*Phalacrocorax auritus*) in the Laurentian Great Lakes [27]. There are not many studies on anthropogenic materials in the nests of inland wetland birds, but noticeable quantities of plastics have been detected in the nests of other birds related to freshwater ecosystems, such as greater thornbirds (*Phacellodomus ruber*) breeding in trees of river floodplain wetlands [28] and chestnut-capped blackbirds (*Chrysomus ruficapillus*) in Brazil [29]. Because of this knowledge gap, Blettler and Wantzen [30] called for scientists to develop more research on the overlooked aspects of plastic pollution, e.g., the entanglement of individuals, the use of plastic as nesting material, and the transport of plastic debris from household waste to terrestrial and freshwater ecosystems. Moreover, Blettler et al. [31] also showed the supremacy of microplastic studies over macroplastic studies, calling again for greater attention to studies related to macroplastics.

In this work, we studied colonies of the ground-nesting gull-billed tern (*Gelochelidon nilotica*) and the black-winged stilt (*Himantopus himantopus*) breeding in inland wetlands to (1) detect the presence of plastics and other anthropogenic materials in their nests; (2) classify the objects detected during searches for predominant patterns; and (3) check for possible damage to adults and chicks. This work was carried out in the La Mancha Húmeda Biosphere Reserve, located in Central Spain, where Gil-Delgado et al. [32] previously found microplastics in the feces of European coots (*Fulica atra*), mallards (*Anas platyrhynchos*), and shelducks (*Tadorna tadorna*). Based on the human presence and intense agriculture in this area, we expected to find anthropogenic debris in the nests of both assessed species.

## 2. Material and Methods

### 2.1. Study Area and Species

We collected data from two temporary inland saline lakes surrounded by an extensive semi-steppe landscape with predominant agricultural use [33]. They are located in the provinces of Cuenca (Manjavacas) and Toledo (Laguna Grande of Quero) in the La Mancha Húmeda Biosphere Reserve, Central Spain (Figure 1A–C). Although most of the lakes are temporary, the natural regime of some is altered by wastewater input, which prolongs the hydroperiod [34,35]. The biosphere reserve wetlands host 26 wader species [36] and are also a wintering ground for cranes (*Grus grus*) [37] and the breeding habitat for other birds such as the gull-billed tern (*Gelochelidon nilotica*) [38], the black-winged stilt (*Himantopus himantopus*), and the greater flamingo (*Phoenicopterus roseus*) [39].

The gull-billed tern is a medium-sized cosmopolitan bird. Its Western Europe population has an Afro-Paleartic distribution with African wintering grounds from which they migrate to breed mainly in European coastal and inland wetlands [40,41]. According to the most recent available information [42], the species is categorized as “Least Concern” on the European Red List of Birds, with a “Stable” status and an estimated population of 16,100–19,700 mature individuals. In our study area, agricultural intensification could be related to a decrease in suitable feeding habitats for this species [43] because it hunts insects in land areas near the wetlands to feed chicks [38]. These colonial breeders build their nests on the ground on islets with clayey bottoms and high vegetation cover that are surrounded by shallow water [44,45] and use sticks, algae, stones, and other debris as material (Figure 1D).

The black-winged stilt is a breeding wader that inhabits European wetlands and winters in southern areas. This species is also listed as “Least Concern” [42] on the European Red List of Birds, and its population is estimated at 101,000–269,000 mature individuals with an increasing trend. They breed in colonies of scattered nests made with plant debris and algae (Figure 1D), mostly located in shallow saline or brackish wetlands with islets above the water level or directly on dry shores [46]. In the study area, this species is among the most common water birds, and it occurs in many of the biosphere reserve lakes [36].

### 2.2. Fieldwork and Analyses

Fieldwork was carried out in July 2022, once all clutches had either hatched or failed and the chicks had left the nests. With the gull-billed tern, we examined 481 and 17 nests in two different colonies located in the Manjavacas saline lake (39°24′56″ N, 2°51′55″ W; Figure 1C). Given the large number of assessed nests and the complex collection of the structure, we analyzed them in situ by collecting all the apparent anthropogenic materials from the nests with tweezers and storing them in individualized zip-lock plastic bags for their subsequent identification.

For the black-winged stilt, we studied 24 nests located at three different points of the Manjavacas saline lake and 4 nests at Laguna Grande de Quero saline lake (39°30′20″ N, 3°14′51″ W; Figure 1C). In this case, whole nests were collected by carefully removing them from the ground with a spatula to avoid adding sticks, stones, and any other materials present below and around each nest. Each nest was independently stored in an individualized plastic waste bag to be later explored in the lab. Here, the aim was to search for anthropogenic materials.

Having obtained synthetic materials, we classified them according to material type: metal, paper, and plastics. Plastics were classified as fragments and threads following Van Franeker et al. [47]. All the identified anthropogenic items were measured with a tape measure and photographed on a white background with a camera. In some cases, the found materials had deformed due to salt water and sun exposure, which prevented them from adopting their original position. For the measurement purposes, we stretched them to restore their actual sizes. We compared the mean size of the debris found in the nests of both species (black-winged stilt n = 4 and gull-billed tern n = 15) through a Welch two-sample t-test carried out with the function “t.test” in R [48]. This test was selected because the Shapiro–Wilk tests (‘shapiro.test‘ function) showed no significant departures from normality for the debris size data (black-winged stilt *p* = 0.860 and gull-billed terns *p* = 0.081), but the Bartlett test (‘bartlett.test‘ function) showed that the debris size data between the species deviated significantly from homogeneity (Bartlett’s K-squared = 11.708, df = 1, *p* < 0.001). All the R functions belong to the R package “stats” [48].

## 3. Results

We confirmed the presence of plastics and other anthropogenic materials in the nests of the two selected species. We found anthropogenic debris in 12 nests of the gull-billed tern (11 in colony A and 1 in colony B; Table 1 and Figure 2), which was 2.4% (12 of 498) of the assessed nests. Most of the debris used as material for nests comprised plastic (fragments or synthetic textile), although we also identified one metallic object and two tissue papers. Except for two cases, we only found one anthropogenic object per nest. The sizes of the found items ranged from 9 mm to 257 mm (Table 1). All the plastic debris was categorized as macroplastics, and all except one (a circular blue object of 9 mm with no clear identification) were fragments that were initially larger, so they could be considered secondary plastics. For colors, we detected no clear pattern, such as white, yellow, blue, black, and green items, which appeared almost equally in the examined nests. We identified the black plastic film fragments (4 and 8 in Table 1 and Figure 2) as debris of a plastic film used in agriculture. The two green plastic filaments (10 and 11 in Table 1 and Figure 2) were fragments of tree protectors used in reforestation and agriculture to prevent herbivores from causing damage. All other detected items could have a domestic origin (i.e., yellow film as part of food wrapper).

Moreover, we found plastic debris in three nests of the black-winged stilt (Table 1 and Figure 3), which represented 12.5% (3 of 24) of all collected nests. All the items detected in the nests of the black-winged stilt could be considered secondary plastics. The small green fragment (1 in Table 1 and Figure 3) detected in the nests of the gull-billed tern could be identified as parts of tree protectors. The others were difficult to identify, but the fragment of polystyrene in nest 3 was probably a part of protective packaging used in agriculture or from a domestic product. We did not detect any entangled, injured, or dead adults or chicks in either species due to the presence of plastic debris in their nests.

In comparison, the items detected in the nests of the black-winged stilt (mean = 12.75 mm, SD = 4.92, n = 4) were significantly smaller (*p* = 0.003, Welch two-sample t-test = 3.641, df = 14.438) than those in the nests of the gull-billed tern (mean = 83.87 mm, SD = 75.04, n = 15; Table 1). The debris sizes were vastly varied, especially those in the nests of the gull-billed tern (see SD for both species and Table 1).

## 4. Discussion

In this study, we confirmed the presence of synthetic materials in the nests of the gull-billed tern and the black-winged stilt that breed in inland salt lakes located in the La Mancha Húmeda Biosphere Reserve. The found items were mostly plastic fragments that resulted from the breakdown of larger objects but were always bigger than 5 mm (macroplastics in accordance with the size classification [47]). According to the type of debris found, we suggest that the birds must have collected them from the surrounding agricultural area (tree protector tubes and agricultural film) and from household garbage disposed around their breeding areas (tissue paper and food wrapper). We detected a difference in size between the anthropogenic materials used by both species but no domination patterns according to their colors. They are two ground-nesting species with similar nest architecture, which could make it difficult to find differences in the selection of nest material. The limited number of nests in which plastic was found makes us cautious about the statistical results. Moreover, a lack of information about the plastic debris availability in the biosphere reserve and its surroundings can make the identification of possible selection patterns difficult. However, given the foraging ranges of the gull-billed terns [38,49], the plastic items in their nests might originate from distant and heterogenous sources. Ecologically different species can act as differential “plastic importers” to inland lakes by increasing the presence of this pollutant at certain points, such as roost areas and protected breeding colonies. They can consequently act as “plastic sinks” in a similar way to what happens in marine ecosystems, where seabirds transport plastics from the sea to deposit them on islets and coastal cliffs [23,50,51].

Due to the scarcity of studies on plastics in wetland bird nests, we cannot determine whether a pattern was occurring in other breeding colonies of these species. However, there is evidence for an active selection of brightly colored plastics by eight seabird species in nest areas of the South Pacific Ocean [18], a preference for white and black debris shown by the brown booby (*Sula leucogaster*) [52], the collection of hard and white items by the Caspian tern (*Hydroprogne caspia*), and the use of greenish fishing gear in the nests of the great cormorant (*Phalacrocorax carbo*) [53]. In our study, both species apparently used plastic and other anthropogenic items as ornaments (most of the nest material comprised sticks and other plant debris, including pebbles and olive pits in many cases). This was perhaps to identify their nests as occupied, to attract potential breeding mates, or to act as a threat to conspecifics to show the dominance of the signaler [54,55,56].

We did not detect any individuals hooked to the nest or showing injuries deriving from entanglement with synthetic materials, although this impact has been confirmed in different species around the world, such as the brown noddy (*Anous stolidus*) [57], the osprey (*Pandion haliaetus*) [58], and the American crow (*Corvus brachyrhynchos*) [59]. In our assessed species, the few synthetic materials that were found and their types may explain the absence of affected birds because we found almost no fibers and ropes (that can entangle legs or wings). In addition, given the chicks’ nidifugous behavior, their presence in the nests was anecdotal and, hence, reduced the probability of being entangled.

## 5. Conclusions

In conclusion, this study is one of the first to show that, beyond ingesting microplastics, wetland-associated birds also use plastics and other synthetic debris as nest materials. As far as we know, this is the first report to confirm the presence of macroplastics in the nests of the gull-billed tern and the black-winged stilt and one of the first to focus on wetland bird nests. In this case, the study site was a protected area (La Mancha Húmeda Biosphere Reserve), which reveals, once again, how anthropogenic debris reaches all kinds of environments independent of their conservation status. Future research is needed to conclude whether the gull-billed tern and the black-winged stilt select certain items from the refuse disposed in both agricultural lands and lakes and to monitor whether the use of anthropogenic debris decreases, remains at its current level, or increases. It would also be important to control if adults or chicks suffer any type of damage, especially if more synthetic threads and ropes appear in nests, which would increase the risk of entanglement. Therefore, it would be important to complete the study into the impact of plastics on these species by analyzing their possible intake, together with the endocrine and immunological effects and consequences for both individuals’ health and survival. Lastly, it would be interesting to assess how the different species that exploit this lake system interact with plastics by transporting them between lakes or from terrestrial to water ecosystems.

## Figures and Tables

**Figure 1 animals-12-03222-f001:**
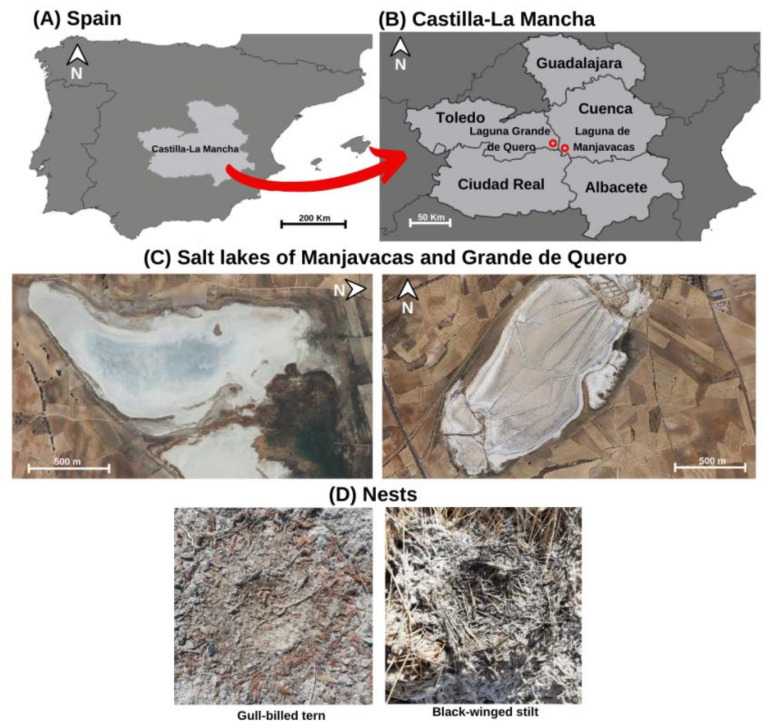
Maps showing the selected study area where the nests of the wetland colonial birds were examined to find plastic debris. (**A**) Castilla-La Mancha, located in Central Spain; (**B**) the two studied salt lakes (Manjavacas and Grande de Quero) located in Cuenca and Toledo; (**C**) aerial view of the studied salt lakes; (**D**) nest details of the two studied species.

**Figure 2 animals-12-03222-f002:**
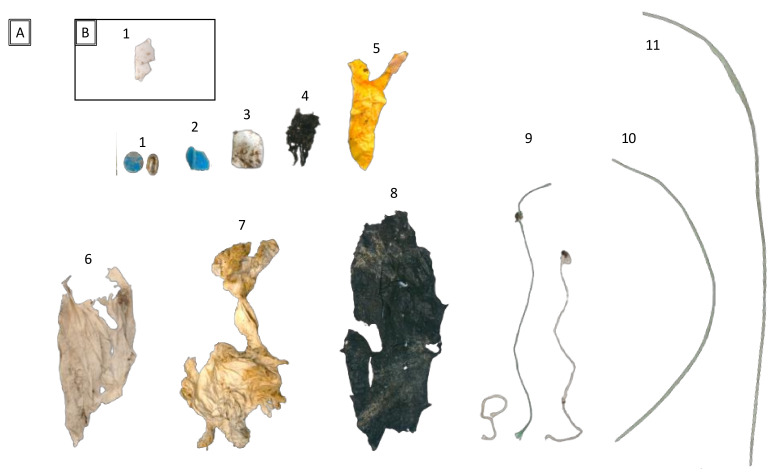
Anthropogenic debris collected in the gull-billed tern (*Gelochelidon nilotica*) nests from two breeding colonies (**A**,**B**) in the La Mancha Húmeda Biosphere Reserve. Each number corresponds to a different nest. Additional information about each item is provided in Table 1.

**Figure 3 animals-12-03222-f003:**
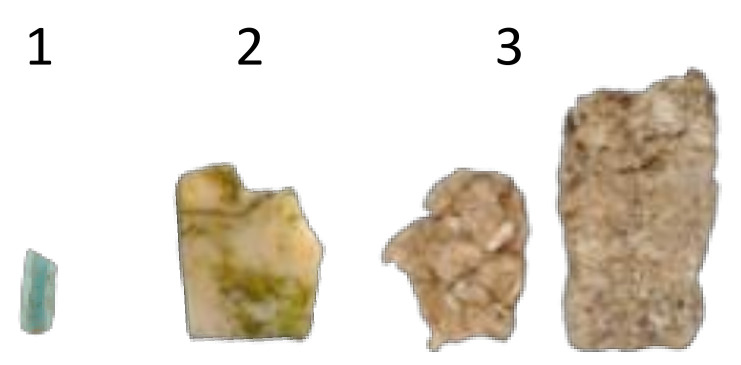
Anthropogenic debris collected in three nests of the black-winged stilt (*Himantopus himantopus*) located in the La Mancha Húmeda Biosphere Reserve. Each number corresponds to a different nest. Additional information about each item is provided in Table 1.

**Table 1 animals-12-03222-t001:** Anthropogenic materials found in the nests of the gull-billed tern (**A**) and the black-winged stilt (**B**) ordered by fragment length, given in mm. * Plastic that was originally white but was observed to be brown due to contact with lake mud. We are not sure what color they were when the birds deposited these fragments in the nests.

	A) Gull-Billed Tern												
	Nest	Item 1	Type	Length (mm)	Color	Item 2	Type	Length (mm)	Color	Item 3	Type	Length (mm)	Color
Colony A	1	undetermined	plastic	9	blue	undetermined	metal	8	gray				
	2	fragment	plastic	10	blue								
	3	fragment	plastic	12	white								
	4	fragment	plastic	30	black								
	5	fragment	plastic	58	yellow								
	6	tissue	paper	83	brown *								
	7	tissue	paper	102	brown *								
	8	fragment	plastic	126	black								
	9	textile	plastic	165	green	textile	plastic	128	white	textile	plastic	64	white
	10	fragment	plastic	179	green								
	11	fragment	plastic	257	green								
Colony B	1	fragment	plastic	27	white								
	**B) Black-Winged Stilt**												
	**Nest**	**Item 1**	**Type**	**Length (mm)**	**Color**	**Item 2**	**Type**	**Length (mm)**	**Color**				
	1	fragment	plastic	7	green								
	2	fragment	plastic	13	white								
	3	fragment	plastic	19	brown *	fragment	plastic	12	brown *				

## Data Availability

Not applicable.

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
