# Peer review of "Plastic Debris in Nests of Two Water Bird Species Breeding on Inland Saline Lakes in a Mediterranean Biosphere Reserve"

_animals, 2022, doi:10.3390/ani12223222_

Round 1

Reviewer 1 Report

Congratulations to the authors for a most worthwhile paper on an understudied ecosystem. There are some minor typos (especially in the References section) to be attended (see attached pdf).

Author Response

Reviewer 1

Many thanks to the reviewer for his/her detailed review and suggestions for a better understanding of the manuscript. After our review, the manuscript has been sent to a native English speaker to improve the writing.

We have also reviewed the statistics and we have done that section again, since as the reviewer noticed there were errors that we had not perceived before.

In this new version we have incorporated almost all the suggested changes, although it has not always been possible to come into conflict with other comments provided by the other reviewers. Similarly, some suggestions were part of text fragments that have been profoundly changed.

We hope this new version of the manuscript will be suitable for publication, although we are open to more changes and suggestions.

Reviewer 2 Report

You have raised a very important point.

My remarks (to be found in the enclosed pdf file) regard language - I suggested some corrections but after the application of them you should send this manuscript to a English native speaker collegue for the final polishing.

The second important thing is the statistical part - remarks also included in the pdf.

Author Response

Many thanks to the reviewer for his/her detailed review and suggestions for a better understanding of the manuscript. After our review, the manuscript has been sent to a native English speaker to improve the writing.

In this new version we have incorporated practically all the suggested changes, although it has not always been possible to come into conflict with other comments provided by the other reviewers. Similarly, some suggestions were part of text fragments that have been profoundly changed.

We hope this new version of the manuscript will be suitable for publication.

Reviewer 3 Report

Well, indeed effect of plastic differ between avian species, also because different nest construction. You mentioned two species, ground nesters, were is not necessary to put a nest construction to a shrub / tree - and it also may produce a big differences. In line 51 you mentioned oculars of plastic and so on, please also check: Antczak, M., Hromada, M., Czechowski, P., Tabor, J., Zabłocki, P., Grzybek, J., & Tryjanowski, P. (2010). A new material for old solutions—the case of plastic string used in Great Grey Shrike nests. Acta ethologica13(2), 87-91.

it is discussion also on plastic availability

Please note that the sample size in your study is so limited, than data have to be used with caution - especially like in the discussion you mentioned on differences between species.

Author Response

We thank the reviewer for his/her comment.

We agree that plastic use could be also influenced by the architecture of the nest and the building process of each species, together with the nest location (ground-tree-shrub) so we added a comment about this in the discussion section.

We read the reference provided by the reviewer, and we agree in the main conclusions of the paper of the great grey shrike. We have added this reference to our introduction section as an additional example of plastic use in nests of a terrestrial bird.

Last, we are aware of the limitations of our sample size, so in the line 214 we start saying that “the limited number of nests in which plastic has been found ( … ) can make the identification of possible selection patterns difficult”. In any case in this new version we use another methodological approach and we remark that we have to be cautious, attending to the limited sample size.

We also sent the new manuscript to an English native speakr that works in the revision of scientific articles.

Reviewer 4 Report

Line 17, 31Stating "we did not detect any pernicious impact on adults or chicks 31 (e.g. individuals entangled, injured or dead)," without mentioning the possible but untested endocrine, immunological and other effects that may reduce recruitment of new animals to the population leaves you open to being misquoted that the plastic has no harmful effects.

line 173, Suggest use the word "length" rather than Longitude, which usually implies geographic position (latitude, longitude)

Line 213 Suggest not "it surround" but rather "it's surroundings"

Line 213, Suggest rather than "can difficult the" either "can complicate the" or "can make the identification of possible selection  patterns difficult"

Line 214/215 "these species can be using these materials randomly, picking up objects that catch their attention."  Items that catch their attention are by definition not random.

line 216 Suggest rather than "items in their nests might proceed from  further and heterogenous origins than" say "items in their nests might originate  from  further and more heterogenous origins than"

Line 229 Plural does not require an "S" Suggest "use of greenish fishing gear in nests"

Line 254 Plural does not require an "S" Suggest "refuse" not "refuses"

In general, it would have been useful to know, where it was possible to be confident,  if the macroplastics were included in the structure of the nest, suggesting material use, or were in faecal material suggesting ingestion.

Author Response

Many thanks to the reviewer for his/her detailed review and suggestions for a better understanding of the manuscript. After our review, the manuscript has been sent to a native English speaker to improve the writing.

Line 17, 31 Stating "we did not detect any pernicious impact on adults or chicks 31 (e.g. individuals entangled, injured or dead)," without mentioning the possible but untested endocrine, immunological and other effects that may reduce recruitment of new animals to the population leaves you open to being misquoted that the plastic has no harmful effects.

Authors: Many thanks for this comment. In this revised version we are added a sentence in the abstract and also one in the conclusion paragraph suggesting the importance of these additional effects on individuals.

Line 173, Suggest use the word "length" rather than Longitude, which usually implies geographic position (latitude, longitude)

Authors: As suggested, in this new version we use length and not Longitude, both in the text but also in the Table 1.

Line 213 Suggest not "it surround" but rather "it's surroundings"

Authors: Done.

Line 213, Suggest rather than "can difficult the" either "can complicate the" or "can make the identification of possible selection  patterns difficult"

Authors: Done.

Line 214/215 "these species can be using these materials randomly, picking up objects that catch their attention."  Items that catch their attention are by definition not random.

Authors: Many thanks for this comment. We changed this sentence to  avoid discussions about the random or intentional selection of certain items.

Line 216 Suggest rather than "items in their nests might proceed from  further and heterogenous origins than" say "items in their nests might originate  from  further and more heterogenous origins than".

Authors: We have changed the sentence as proposed by the reviewer.

Line 229 Plural does not require an "S" Suggest "use of greenish fishing gear in nests"

Authors: Many thanks, we changed the sentence using gear and not gears.

Line 254 Plural does not require an "S" Suggest "refuse" not "refuses"

Authors: In this new version we use refuse.

Reviewer: In general, it would have been useful to know, where it was possible to be confident,  if the macroplastics were included in the structure of the nest, suggesting material use, or were in faecal material suggesting ingestion.

Authors: This comment is very interesting. In our case, we did not detect feces. The plastic found in all cases was found inside the nests of these birds, which were analyzed in situ or in the laboratory. Furthermore, the size and shape of some plastics found would make it difficult to ingest by these species. In any case, although in this occasion we have not collected pellets or feces it could be interesting to carry out a parallel project including them, to see if they contain plastics and to see if they resemble those found in nests or have other sizes and colors, which could indicate a differential selection depending on whether they ingest it or use it as nest material. In the same lagoon system, microplastics have been found in the feces of other aquatic birds, and one of co-authors of this work is developing research to find microplastics in the feces of more birds in the area.

Round 2

Reviewer 2 Report

You have greatly improved the manuscript.

Reviewer 3 Report

I focussed mainly on my previous comments and the are solved